# Essential Oils as a Good Weapon against Drug-Resistant *Candida auris*

**DOI:** 10.3390/antibiotics11070977

**Published:** 2022-07-20

**Authors:** Liliana Fernandes, Rita Ribeiro, Raquel Costa, Mariana Henriques, M. Elisa Rodrigues

**Affiliations:** 1Centre of Biological Engineering, LMaS—Laboratório de Microbiologia Aplicada à Saúde, Campus de Gualtar, University of Minho, 4710-057 Braga, Portugal; lilianafernandes@ceb.uminho.pt (L.F.); ritammr23@gmail.com (R.R.); elisarodrigues@ceb.uminho.pt (M.E.R.); 2LABBELS–Associate Laboratory, Campus de Gualtar, University of Minho, 4710-057 Braga, Portugal; 3Aromas Aqua Spa–Clínica Saúde, Praça 5 Outubro nº 32, 4730-731 Braga, Portugal; raqcosta@sapo.pt

**Keywords:** *Candida auris*, biofilm, resistant species, natural therapy, phytotherapeutic applications, essential oils

## Abstract

*Candida auris* is a recently found *Candida* species, mainly associated with nosocomial outbreaks in intensive care hospital settings, and unlike other *Candida* species, it can be transmitted through person-to-person or by contact with surfaces. *C. auris* is described as resistant to first-line antifungals and, consequently, associated with high mortality. Nowadays, essential oils (EOs) are known to be effective against fungal and bacterial infections. This work aimed to evaluate the effect of four EOs (tea tree, niaouli, white thyme and cajeput) against *C. auris*. The EO’s effect on *C. auris* planktonic growth was evaluated by the minimum inhibitory concentration determination and by the agar disc diffusion method. Then, the same effect was evaluated on biofilm by colony-forming units’ enumeration. The results showed that EOs were able to inhibit the *C. auris* planktonic growth, with an MIC50 between 0.78 and 1.56% and halos of 20–21 mm for white thyme and tea tree and 13–14 mm for cajeput and niaouli. In addition, the EOs were also able to completely inhibit biofilm formation. Moreover, white thyme and cajeput completely eradicate pre-formed biofilms, while tea tree and niaouli significantly reduce it. Thus, this work demonstrates that EOs are a possible therapeutic alternative and a future perspective for the hard fight against *C. auris*.

## 1. Introduction

*Candida auris* is a new *Candida* species recently found after being isolated from the external auditory canal of a Japanese patient, and since then, it has been found and reported in more than 6 continents and 30 countries [1,2,3]. *C. auris* is mainly associated with nosocomial outbreaks in intensive care settings, where it has been recovered from the skin of healthcare workers and patients, areas that have been in contact with infected patients and a wide variety of surfaces [4]. *C. auris* has shown high colonisation and survival capabilities and, unlike the majority of the other *Candida* species, can be transmitted through person-to-person or through contact with surfaces or contaminated equipment [4]. This yeast causes highly invasive infections with a high mortality rate in hospitalised patients, with immunosuppression and underlying chronic disease being the main risk factors [5]. In fact, several virulence factors commonly associated with *Candida albicans* (considered the most virulent member of the genus) have been identified in *C. auris*, such as phospholipase and proteinase activities and biofilm formation, although *C. auris* isolates produce only rudimentary pseudohyphae [6]. In addition, *C. auris* has already been described as resistant to the first-line antifungal agent, fluconazole (FLC), exhibiting variable susceptibility to other azoles and echinocandins [1]. Thus, the search for alternative therapies from natural sources featuring new mechanisms of action combined with new strategies has become crucial. Nowadays, it is well known that some plant products, such as essential oils (EOs), are effective against fungal and bacterial infections, presenting good bioactivity and low toxicity [7]. The strong antimicrobial activities of EOs and their components are described in several studies [8,9,10,11]. EOs have the ability to hinder the growth of pathogens, increasing membrane permeability, disrupting the cell membrane, inducing leakage of vital intracellular constituents and interrupting cell metabolism and enzymatic kinetics [10]. Tran et al. [11] demonstrated for the first time the potential of EO from *Cinnamomum zeylanicum* to exert antifungal activity against planktonic *C. auris* cells, highlighting their ability to damage the membranous structures of fungal cells and haemolytic activities. In addition, it was also demonstrated that EO arborvitae led to a significant decrease in the intrinsic planktonic growth rate of *C. auris*. However, in general, there are very few studies on the effect of EOs on *C. auris,* especially on *C. auris* biofilm. In this study, four EOs were evaluated; the choice of these four specific EOs was based on clinical evidence provided by a clinic (Costa Raquel, Aromas Aqua Spa–Health Clinic) and due to their characteristics. Tea tree EO is recognised for its antifungal, antiseptic, anti-inflammatory and anti-infective properties; it also stimulates the immune system and has a calming effect. Cajeput and white thyme EOs are recognised for their respiratory tract cleansing properties and are used for their calming properties. Niaouli EO is an expectorant and decongestant that strengthens the immune system and can be used for skin problems such as acne, itching, dermatitis or sunburn [12].

Therefore, the aim of this work was to evaluate the effect of tea tree, cajeput, niaouli and white thyme EOs against *C. auris* species.

## 2. Results

### 2.1. Essential Oils Composition

The identified compounds and their relative contents are listed in Table 1 according to their percentage ascending on a non-polar column. The major compounds of the EOs are cineole (65%), α-terpineol (11%) and linalool (3%) in cajeput oil; p-cymene, limonene and 1,8-cineole (57%) in niaouli oil; terpinen-4-ol (42%), γ-terpinene (21%) and α-terpinene (10%) in tea tree oil; and borneol (32%), α-terpineol (16%) and carvacrol (9%) in white thyme oil.

### 2.2. Antifungal Susceptibility Testing

#### 2.2.1. Fluconazole

To verify *C. auris* resistance to FLC, both minimum inhibitory concentrations (MIC_50_) and minimum fungicidal concentrations (MFC) were firstly evaluated. The results showed an MIC_50_ of 125 µg/mL and an MFC of ≥125 µg/mL.

#### 2.2.2. Essential Oils

##### Planktonic Antimicrobial Susceptibilities

The MIC_50_ and MFC of white thyme, tea tree, cajeput and niaouli EOs against planktonic populations of *C. auris* were determined. A lower MIC_50_ was observed for tea tree EO (0.78% (*v*/*v*)) in comparison to the other EOs tested, with an MIC_50_ value of 1.56% (*v*/*v*). Similarly, the MFC recorded for the tea tree EO was lower (1.56% (*w*/*v*)) than for white thyme, cajeput and niaouli EOs (3.12% (*w*/*v*)).

The antifungal activity of EOs (white thyme, tea tree, cajeput and niaouli) against *C. auris* was also evaluated using the agar disc diffusion method, and the results are summa-rised in Figure 1. As a negative control, 70% ethanol (*v*/*v*) was used, resulting in a 9 mm halo. Regarding the effect of EOs, both white thyme (21.4 ± 0.5 mm) and tea tree (20.0 ± 1.9 mm) oils demonstrated stronger antifungal activity against the tested strains when compared with cajeput (13.8 ± 1.1 mm) and niaouli (13.3 ± 1.1 mm) oils. Furthermore, despite similar diameters between the tea tree and white thyme oil, there was a marked decrease in the *C. auris* biomass all over the petri dish with the application of white thyme EOs (Figure 1).

##### Biofilms Antimicrobial Susceptibilities

The antifungal effect of white thyme, tea tree, cajeput and niaouli EOs in biofilm formation and 24 h old biofilms of *C. auris* was evaluated (Figure 2). Direct application of two drops (2.4% (*v*/*v*)) of EOs on biofilm formation (Figure 2A) led to a total inhibition of biofilm growth. Moreover, the oils were also applied in pre-formed biofilms (Figure 2B), and it was observed that the application of two drops (2.4% (*v*/*v*)) of white thyme and cajeput oils induced a total eradication of the 24 h old biofilms. In addition, tea tree and niaouli oils significantly reduced viable cells of those pre-formed biofilms (*p* < 0.0001), with reductions of 5 Log_10_ CFU/mL and 4 Log_10_ CFU/mL (*p* < 0.001), respectively.

## 3. Discussion

The general resistance of *Candida* spp. to most available antifungal drugs and the emergence of multidrug-resistant species, such as *C. auris*, make it urgent to develop new drugs or more effective therapies. Therefore, this study aimed to evaluate the antifungal activity of four EOs against *C. auris* in both planktonic cells and biofilms (on its formation and on pre-formed biofilms).

The results of antifungal susceptibility test (MIC_50_ of 125 µg/mL and a MFC of ≥ 125 µ/mL) confirm, that, as expected, *C. auris* is highly resistant to FLC (sensitive < 32 µg/mL; resistant ≥ 32 µg/mL) [13]. Recently, genetic analyses of the *C. auris* species have reported mutations in genes related to the development of triazole resistance. In addition, the presence of genes encoding proteins involved in the acquisition of resistance, such as protein kinases, efflux pumps and major facilitator superfamilies, have also been reported [14]. Thus, considering the still increasing problem of *C. auris* drug resistance, the antifungal properties of the EOs can be considered a promising alternative.

Regarding the antifungal activity of EOs against planktonic *C. auris* cells, all the EOs were able to inhibit the growth of *C. auris*, with MIC_50_ values between 0.78% (*v*/*v*)−1.56% (*v*/*v*) and EOs MFC was 2 × MIC (Figure 1). The four EOs tested showed inhibitory and fungicidal properties at low concentrations. Tran et al. [11], one of the few studies that present the MIC and MFC values of EOs in *C. auris*, also verified very low MIC and MFC values for cinnamon EO of < 0.03–0.13% (*v*/*v*) and 0.25% (*v*/*v*), respectively.

In the agar disc diffusion method, white thyme and tea tree oils demonstrated stronger antifungal activity when compared with cajeput and niaouli oils (Figure 1). Our results are in line with several reports of high antifungal activity of thyme and tea tree oils against both susceptible and drug-resistant strains of various *Candida* species [15,16]. Indeed, EOs and their constituents have been used against a wide range of fungal pathogens since they have the ability to hinder the growth and development of a diverse range of pathogens [10]. In fact, according to a previous study, although the efficiency of EOs differs substantially between *Thymus* species, EOs from *Thymus* species can be pointed out as a great contribution to the treatment of *C. auris* infections [17].

After these good preliminary results, the antifungal effect of EOs in biofilm formation and 24 h old biofilms of *C. auris* was evaluated (Figure 2). Our results confirm the high ability of *C. auris* to form biofilms (Figure 2—(+) control), as previously described by Horton et al. [18]. These authors demonstrated that the growth in synthetic sweat medium and in the skin of pigs allows *C. auris* to form dense biofilms that resist desiccation and thrive in conditions of evaporation. In addition, the draft genome that identifies several proteins involved in biofilm formation and recent descriptions of aggregative and non-aggregative phenotypes indicate the possibility of heterogeneous *C. auris* biofilm formation [6].

In relation to the application of EOs on biofilm formation (Figure 2A), it was possible to verify that the application of only two drops (2.4% (*v*/*v*)) of oil (white thyme, tea tree, cajeput or niaouli) induced to a total inhibition of biofilm growth after 24 h. Furthermore, the application of the same volume of white thyme and cajeput oils led to the total eradication of 24 h old biofilms (pre-formed biofilms) (Figure 2B), showcasing their great efficacy. Besides, tea tree and niaouli oils significantly reduce viable cells of those pre-formed biofilms (*p* < 0.001). In fact, the good anti-*Candida* activity of thyme was also verified in many reports [17,19]. Similar to the results obtained in this study, Asdadi et al. showed that EO of *Thymus satureiodes* (white thyme oil) was able to inhibit the growth of non-*Candida albicans* species (*Candida krusei*, *Candida dubliniensis* and *Candida glabrata*) resistant to conventional antifungals, such as FLC and amphotericine B [19].

In this study, the major compounds of the white thyme oil are borneol, α-terpineol and carvacrol. It has been demonstrated that oxygenated terpene compounds, such as carvacrol, are often considered the main compounds responsible for modifying membrane permeability by chemical reaction with amino and hydroxylamine groups of membrane proteins [8]. Indeed, carvacrol has the highest antifungal activity, inhibiting the formation of hyphae and biofilms in *Candida* species [5,20]. Furthermore, terpinen-4-ol and α-terpineol are recognised as potent compounds with a fungicidal effect [16]. Thus, the very good outcomes obtained in this work are justified by the important antifungal role of these compounds.

In turn, the major compounds of the tea tree EO are terpinen-4-ol, γ-terpinene and α-terpinene. Tea tree oil and its components increase yeast cell permeability and membrane fluidity and become embedded in the lipid bilayer membrane, eventually disrupting its structure [21]. Besides that, tea tree oil also inhibits the formation of germ tubes or mycelial conversion and inhibits respiration in *C. albicans* [21]. In addition, Mondello et al. [16] suggested that terpinen-4-ol is a likely mediator of the in vitro and in vivo activity of tea tree oil and that it could control *C. albicans* vaginal infection. The EO of *Melaleuca quinquenervia* (niaouli oil) was found to be effective against bacteria as well as a range of fungi, including *C. albicans* [22]. Additionally, Keereedach et al. found that *Melaleuca cajuputi* (cajeput oil) had potent antifungal activity against clinical isolates of *C. albicans* resistant to FLC, where it was able to reduce the MIC of FLC and reduce the expression level of MDR1 (an important gene that plays a role in resistance to azole drugs in *C. albicans*) [23]. However, so far, there are few studies regarding the effect of niaouli and cajeput oils in *Candida* species, with this being the first study to use both oils in *Candida* biofilms. The major compounds of these two oils are cineole, α-terpineol and linalool in cajeput oil and ρ-cymene, limonene and 1,8-cineole in niaouli oil. 1,8-Cineole showed some anti-*Candida* activity with a fungicidal effect [24]. In addition, linalool exerts antifungal activity by disrupting the membrane integrity and interrupting the cell cycle of planktonic *C. albicans*, inhibits germ tube formation and exhibits antifungal activity against *C. albicans* cells in biofilms. The inhibition of hyphae induction by limonene at low concentrations may be responsible for the inhibition of biofilm formation. However, ρ-cymene, a thymol precursor, is generally defined as inactive [25,26].

Nevertheless, it is complicated to point the antifungal activity of a complex mixture (EOs) to specific constituents. In fact, it is believed that the antifungal activity most probably results from the combined effect of different compounds on several cellular targets [27]. In this study, the composition of the tested EOs suggests that each oil is featured by components common to the four EOs (27–35% of the total composition) and specific components (19–33%). Thus, it is reasonable to speculate that the antifungal activity of these four EOs can also be related to the presence of specific compounds, such as terpinen-4-ol, carvacrol, α-terpineol, linalool and limonene. Studies report that terpinen-4-ol, a major component of tea tree oil (42%), has a lipophilic characteristic, probably presenting a direct action on the cell membrane structure and associated enzymes [28]. Carvacrol, present in white thyme oil (9%), and linalool, present in the four EOs under study in a range between 0.2–3%, are associated with the modification of membrane permeability and inhibition of hyphae and biofilm formation [8]. The antifungal activity of α-terpineol, present in the four EOs (3–16%), also occurs through the disruption of cell walls and cytoplasm, resulting in abnormal hyphae [29]. The limonene constituent of tea tree, niaouli and white thyme oils act on the genetic material of yeast, leading to damage to the cell wall and intracellular structures, including nuclear alterations (condensation of genetic material and specific changes in mitochondria). This compound also induces dramatic structural changes in organelles, accompanied by cell wall disruption [30]. However, to better understand the antifungal effect of these EOs, the synergistic or antagonistic effect between the most abundant compounds and also those present in a smaller percentage in the mixture should be investigated [31]. Thus, more studies are needed in order to understand the interactions between components and, consequently, the mechanisms of action of EOs.

## 4. Materials and Methods

### 4.1. Essential Oils

This study evaluated the antifungal activity of fours EOs, namely cajeput (*Melaleuca cajaputi* (leaf); florame^®^ (Lot 801736), Portugal); niaouli (*Melaleuca quinquenervia* (leaf); florame^®^ (Lot 100378), Portugal); tea tree (*Melaleuca alternifolia* (leaf); florame^®^ (Lot 903025), Portugal) and white thyme (*Thymus satureiodes* (flowering tops); florame^®^ (Lot 800180), Portugal) (all with 100% purity). All EO samples were stored wrapped in aluminium foil in order to protect from light at room temperature.

Analysis of the EOs was carried out by florame^®^ (Saint-Rémy-de-Provence, France) (Appendix A), using Gas Chromatography (GC) with Flame-Ionisation Detection (FID) under the following conditions: the hydrogen carrier gas; the polar Elite-WAX column (100% polyethylene glycol) (60 m/0.25 mm/0.25 μm) and the non-polar Elite-5 columns (5% diphenyl, 95% dimethylpolysiloxane) (60 m/0.25 mm/0.25 μm) for tea tree, white thyme and cajeput EOs. Analysis of the niaouli EO was carried out using a GC 6890 MS 5975 instrument at the following conditions: the injected sample volume was 1 μL; the helium carrier gas flow rate was 1 mL min^−1^; an HP5 ms capillary column (30 m/0.25 mm) was used, with a film thickness of 0.25 μm; the column temperature was 60 °C increasing at 2 °C min^−1^ to 250 °C; and the mass range was 40–450 m z^−1^.

### 4.2. Microorganisms and Culture Conditions in This Study

*Candida auris* NCPF 8971 was stored at −80 ± 2 °C in Sabouraud Dextrose Broth (SDB; Liofilchem, Roseto degli Abruzzi, Italy) with 20% (*v*/*v*) glycerol. Before each assay, *C. auris* was subcultured onto Sabouraud Dextrose Agar (SDA; Liofilchem) and incubated aerobically at 37 °C for 18–24 h. SDA plates were prepared from SDB supplemented with 2% (*w*/*v*) agar (Liofilchem).

### 4.3. Antifungal Susceptibility Testing

#### 4.3.1. Fluconazole

MICs and MFCs of FLC were determined using the broth microdilution method according to Clinical and Laboratory Standards Institute (CLSI) guidelines (CLSI M27-A4), with some modifications [32]. Briefly, the initial cell concentration for *C. auris* was adjusted for 2 × 10^5^ Colony-Forming Units (CFU)/mL in RPMI 1640 and dispensed into 96-well round bottom microtiter plates with the FLC solution, both in a proportion of 1:2. Positive (*C. auris* suspension) and negative (broth medium RPMI 1640) controls were included. All microtiter plates were incubated for 24–48 h at 37 °C. MIC was obtained by visual observation of the turbidity gradient and by the determination of the optical density at 570 nm. MIC_50_ endpoints were established as the lowest concentration of FLC that resulted in a decrease of growth by ≥ 50% relative to the positive control (*C. auris* suspension). To determine MFC, each well was sub-cultured onto SDA plates and incubated for 24 h at 37 °C. MFC endpoints were defined as the concentration in the first well with no growth on the SDA plate. The experiment was performed in three independent assays in triplicate.

#### 4.3.2. Essential Oils

##### Planktonic Antimicrobial Susceptibilities

MICs and MFCs of cajeput, niaouli, tea tree and white thyme EOs were determined as described above, with some modifications due to the use of volatile compounds. After the concentration was adjusted, the cell suspension was dispensed into glass wells into glass Petri dishes (1 mL/well) with the EOs diluted (12.5% (*v*/*v*)–0.02% (*v*/*v*)) in vegetable oil and almond oil (*Prunus amygdalus*, AB—Agriculture Biologique, Paris, France), with both at a ratio of 1:2. Positive (*C. auris* suspension with almond oil) and negative (RPMI broth medium) controls were included. The set glass wells into glass Petri dishes were incubated for 48 h at 37 °C. MIC_50_ and MFC were determined as described above [17].

The agar disc diffusion method described by Tran et al. and Touati et al. [11,33], with some modifications, was used for the determination of the antifungal activities of cajeput, niaouli, tea tree and white thyme oils. Briefly, a swab dipped in cell suspensions (pre-inocula) adjusted to 1 × 10^8^ cells/mL was spread onto SDA plates. Then, sterile 6 mm blank disks (Liofilchem) impregnated with 25 µL of each oil (100% (*v*/*v*)) were placed over the agar surface, and plates were incubated at 37 °C for 24 h. Plates with disks loaded with ethanol 70% (*v*/*v*) and almond oil were also included as positive and negative controls, respectively. The inhibition zones induced by different EOs were measured (mm).

Both experiments were performed in triplicate and in three independent assays.

##### Biofilm Antimicrobial Susceptibilities

*C. auris* biofilms were developed as described by Stepanovi’c et al. and Ribeiro et al. [17,34], with some modifications due to the use of volatile compounds. Briefly, pre-inocula (pure liquid cultures) of *C. auris* were maintained in SDB for 18 h at 37 °C under agitation (120 rev/min). Then, the initial cell concentration was adjusted for 1 × 10^5^ cells/mL. The cellular suspension was transferred, under aseptic conditions, to glass wells inside glass Petri plates (1 mL/well). The *C. auris* biofilm culture was incubated aerobically for 24 h on a horizontal shaker at 120 rpm and 37 °C. The effect of four EOs (100%) (tea tree, niaouli, white thyme and cajeput) was evaluated on biofilm formation and on 24 h old biofilms. For this, two drops (25 µL with a final concentration of 2.4% (*v*/*v*)) of each oil were added to the wells containing a cell suspension (biofilm formation) or pre-formed biofilms (24 h old biofilms). Positive (*C. auris* suspension) and negative (SDB broth medium) controls were included. All plates were then incubated in the same conditions. After 24 h, the wells were washed with saline solution to discard the planktonic fraction, and the microdrop technique was used. Biofilm cell suspensions were serially diluted in saline and plated on SDA plates (incubated aerobically for 24 h at 37 °C), and the cultivable cells were enumerated. Values of cultivable sessile cells were expressed as Log CFU per ml (Log_10_ (CFU/mL)). The biofilm experiment was performed in triplicate with three independent assays.

### 4.4. Statistical Analysis

Biofilm antimicrobial susceptibility results were analysed using the Prism software package (GraphPad Software version 6.01 for Macintosh). Two-way ANOVA tests were performed, and means were compared by applying Tukey’s multiple comparison test. Statistically significant differences were considered significant when *p* < 0.05. At least three independent experiments were performed (in triplicate).

## 5. Conclusions

*C. auris* is recognised as a notorious nosocomial pathogen that requires urgent efforts to develop more efficient and safer alternative treatments due to high transmissibility, incorrect use of antifungal drugs, identification challenges and consequent treatment failures. In this sense, EOs are an alternative therapy for this important pathogen. The results of this study demonstrate a high efficacy of the four EOs tested to inhibit the planktonic growth and completely inhibit *C. auris* biofilm formation. Altogether, the data suggest that the application of these four EOs, particularly white thyme and cajeput, is an efficient alternative for preventing and treating infections caused by *C. auris* biofilms and also reduces dependence on existing conventional antimicrobials.

## Figures and Tables

**Figure 1 antibiotics-11-00977-f001:**
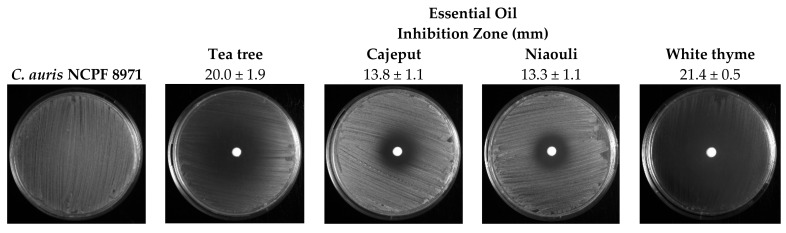
Antifungal activity of essential oils on *C. auris* NCPF 8971 evaluated using the disk diffusion assay.

**Figure 2 antibiotics-11-00977-f002:**
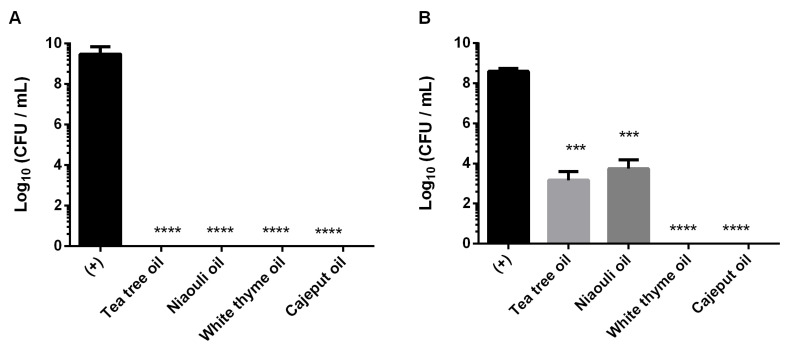
Antifungal effect of four essential oils (tea tree, niaouli, white thyme and cajeput) in biofilm formation (**A**) and 24 h old biofilms (**B**) of *C. auris*. *** *p* < 0.001, **** *p* < 0.0001 indicates a statistically different reduction in comparison with the control (+).

**Table 1 antibiotics-11-00977-t001:** Identification of the components of tea tree, cajeput, niaouli and white thyme essential oils expressed as a percentage (%) of individual compounds in the total essential oil sample.

Tea Tree Oil		Cajeput Oil		Niaouli Oil		White Thyme Oil	
Compounds	(%)	Compounds	(%)	Compounds	(%)	Compounds	(%)
Terpinen-4-ol	42.20	Cineole	64.83	p-Cymene +Limonene+1,8-cineole	57.14	Borneol	31.86
γ-terpinene	21.38	α-terpineol	11.19	α-pinene	12.00	α-terpineol	15.95
α-terpinene	10.19	Linalool	3.21	α-terpineol	8.55	Carvacrol	8.63
Terpinolene	3.46	Myrcene	1.81	Viridiflorol	7.29	Camphene	7.26
α-terpineol	3.25	α-pinene	1.71	β-pinene	3.46	trans-β-Caryophyllene	6.62
1,8-cineole	3.03	γ-terpinene	1.71	β-Caryophyllene	1.22	Bornyl acetate + Thymol	4.54
α-pinene	2.52	Terpinolene	1.31	Ledene	1.19	α-pinene	4.12
p-Cymene	1.49	β-selinene	1.28	γ-terpinene	1.04	Linalool	2.92
Limonene	1.08	Sabinene	1.23	Terpinen-4-ol	1.01	p-Cymene	2.14
Aromadendrene	1.00	α-caryophyllene	0.92	Nerolidol	0.73	Terpinen-4-ol	1.85
Bicyclogermacrene	0.85	Terpinen-4-ol	0.91	Ledol	0.73	γ-terpinene	1.83
Ledene	0.82	α-selinene	0.89	β-myrcene	0.56	Camphor	1.27
δ-cadinene	0.78	Caryophyllene oxide	0.89	Alloaromadendrene	0.38	Limonene	1.07
Myrcene	0.72	β-eudesmol	0.57	Terpinolene	0.34	Thymol methyl ether	0.82
α-thujene	0.71	α-eudesmol	0.55	α-terpinene	0.26	β-pinene	0.78
β-phellandrene	0.68	α-terpinene	0.54	Linalool	0.24	1,8-cineole	0.73
β-pinene	0.66	p-Cymene	0.43	Benzaldehyde	0.22	α-terpinene	0.49
Sabinene	0.47	Germacrene D	0.33	α-Caryophyllene	0.20	y-cadinene	0.47
α-phellandrene	0.37	α-phellandrene	0.26	β-selinene	0.20	δ-cadinene	0.44
Linalool	0.36	Borneol	0.25	y-cadinene	0.18	Terpinolene	0.41
Alloaromadendrene	0.32	Geranyl acetate	0.25	Camphene	0.18	Caryophyllene oxide	0.36
Globulol	0.29	α-thujene	0.20	δ-cadinene	0.16	Myrcene	0.35
α-gurgujene	0.25	Benzaldehyde	0.16	Isopulegol	0.13	Tricyclene	0.33
trans-β-Caryophyllene	0.25	10-epi-gamma-eudesmol	0.10	α-phellandrene	0.11	α-thujene	0.25
Viridiflorol	0.24	α-Copaene	0.06	α-thujene	0.11	Ethanol	0.17
α-Copaene	0.10			Citronellol	0.07		
				Alcool fenchylique	0.07		
				Borneol	0.06		
				Patchoulene	0.06		
				Methyl benzoate	0.05		
				trans-β-ocimene	0.04		

## Data Availability

All data generated or analysed during this study are included in this published article.

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
