# Peer review of "Essential Oils as a Good Weapon against Drug-Resistant Candida auris"

_antibiotics, 2022, doi:10.3390/antibiotics11070977_

Round 1

Reviewer 1 Report

The authors present an interesting study looking at the susceptibility of C. auris to a series of essential oils. The threat of drug-resistant C. auris is a major concern and a hot topic of research in this field at the moment. With that, although the study is well-designed, there are a number of already published reports that provide almost the same information as this manuscript, they just use other plants related to those used in this study (Garden thyme vs. White thyme), or target the same mechanisms in C. albicans. The major novelty in this work is then the HPLC data that provides a list of possible chemicals that can be tested independently in the future for their anti-fungal properties. The current version requires some additional data setting this study apart from existing reports.

I did also have one comment regarding formatting. Table 1 should really be Figure 1 as it contains images.

Author Response

The authors appreciate the reviewer’s comment and the manuscript was improved. Also, the formations corrections were performed as recommended (table 1 was legend as Figure 1).

Reviewer 2 Report

lINE NO: 246,

The experimental design in this study is very straightforward.

The introduction part is well discussed.

MIC and MFC - should be expanded in their first-time usage.

typo errors should be corrected throughout the manuscript: ex., 37 oC.

For the biofilm assay - it is better to include the 96 well plate - culture condition - image.

Author Response

The authors appreciate the reviewer's comment and consider that even though the experimental design is straightforward the study present novelty and the results are of extreme importance in the field. The authors are also grateful for the positive comment on the discussion of the introduction. Additionally, the corrections required were made in the manuscript.

Furthermore, we included the image here for your elucidation (figure 1 - in attachment), but if you prefer, we can introduce it in the revised manuscript.

Reviewer 3 Report

C. auris is a nosocomial pathogen that needs urgent efforts to search for alternative therapeutic treatments, due to its high transmissibility and the spread of resistant strains. In this scenario, the authors described the EOs effect anti-C.auris. Results indicated a strong potential of four EOs in inhibiting the planktonic growth and biofilm formation. The article is well organized and properly written. I have some suggestions for the authors:

- Change throughout the text mL-1 with /mL;

- Have these compounds an antiviral activity?

- Is not more simply for the reader to observe graphs and not plates to show anti-C.auris activity?

- Which are the controls used in the test?

Author Response

The corrections required was made in the manuscript. In general, it is demonstrated in the literature that essential oils have shown promise as antiviral agents against various pathogenic viruses, including influenza and other respiratory viral infections[1]. Indeed, Garozzo et al. show that tea tree essential oil has antiviral activity against the influenza virus and that the antiviral activity was mainly attributed to terpinen-4-ol, the main active component [2]. In fact, this compound, terpinen-4-ol, is present in the four essential oils tested in this work.

The authors believe that it is usually more appreciative visual images for MIC, in addition to the detailed information (inhibition zone (mm)).

In the antifungal susceptibility test to fluconazole, the positive control is the suspension of C. auris and the negative control is RPMI 1640 medium. In the antifungal susceptibility test to the essential oil, the negative control is the suspension of C. auris with almond oil (vegetable oil used to dilute essential oil) applied in the same condition as the assay and the ethanol 70 % (v/v) as a positive control. In the test of biofilm antimicrobial susceptibilities, the positive control was the C. auris biofilm and the negative control was the SDB broth medium. The information of the control conditions was clarified in the text in the materials and methods section.

  1. da Silva, J.K.R.; Figueiredo, P.L.B.; Byler, K.G.; Setzer, W.N. Essential Oils as Antiviral Agents, Potential of Essential Oils to Treat SARS-CoV-2 Infection: An In-Silico Investigation. International Journal of Molecular Sciences 2020, 21, doi:10.3390/IJMS21103426.
  2. Garozzo, A.; Timpanaro, R.; Bisignano, B.; Furneri, P.M.; Bisignano, G.; Castro, A. In Vitro Antiviral Activity of Melaleuca Alternifolia Essential Oil. Lett Appl Microbiol 2009, 49, 806–808, doi:10.1111/J.1472-765X.2009.02740.X.

Reviewer 4 Report

The paper ‘’Essential oils as a good weapon against drug-resistant Candida auris’’ reports interesting subject regarding the antimicrobial effect of essential oil regarding the resistant candida strain. However, I am really sorry to say that the present paper must be rejected, as the analysis of the essential oil is incorrect. KI or AI were not calculated and compared with those published in the literature. Without these values ​​the analysis can have a large% of poorly characterized components. This calculation is mandatory in the analysis of the constituents of essential oils. See: R. P. Adams ; Identification of Essential Oil Components By Gas Chromatography / Mass Spectrometry, 4th Edition 2007.

In addition, the manuscript is not well structured and present many scientific gaps.

As examples:

Why did you use two different methods to analyse the essential oils ? ! GC-FID for tea three, white thyme and cajeput and GC/MS for niaouli !!!. And how did you identify the listed compound of each oil?

Line 234 : Table 2 must be in the result section

Line 275 : I think a volume of 25 µL is to much to be impregnated in disc of 6 mm. How did you apply this volume ??

Line 290 : Why did you use the concentration 2.4 % (v/v) exactly ??

Author Response

We would like to thank you all the points raised by the reviewer. In fact, as these oils were already used in clinical practice and they were selected according to the clinician experience, we used the composition given by each supplier, therefore we did not calculate the RI and AI as proposed. Additionally, regarding the volume used for the halo test, it was based on Touati et al.. Moreover, the concentration of 2.4% is equivalent to the application of 25 µl (same volume applied in the planktonic antimicrobial susceptibilities test) in 1 ml of cell suspension which results in 2.4% of the total volume in the well.

The authors hope that, by responding to individual reviewers' comments, they have clarified the doubts raised by the reviewer 4. In addition, the article was improved in general, taking into account all the suggestions and comments of four more reviewers who considered the article innovative, with valid methods and with results relevant to the field.

Touati, I.; Ruiz, N.; Thomas, O.; Druzhinina, I.S.; Atanasova, L.; Tabbene, O.; Elkahoui, S.; Benzekri, R.; Bouslama, L.; Pouchus, Y.F.; et al. Hyporientalin A, an Anti-Candida Peptaibol from a Marine Trichoderma Orientale. World Journal of Microbiology and Biotechnology 2018, 34, 1–12, doi:10.1007/S11274-018-2482-Z/TABLES/5.

Reviewer 5 Report

The manuscript gives an insight in the antifungal properties of four selected essential oils against drug resistant Candida auris specie. The rationale for the establishment of this research was nicely founded. This ascertains its novelty and the importance.

Nevertheless, a few substantial allegations could be addressed.

Line 26

In the opinion of the Reviewer the context of clinical application the world aromatherapy should be not used, as it is a part of alternative, not evidence based medicine. The EOs are used of course in the officially approved phytotherapeutic applications.

Line 23-24

This sentence is an exaggeration. This work demonstrates the antifungal activity od EO’s, but we can not predict how they will act after the application to the patient. The therapy is a future prospect and the results presented in this manuscript suggests such possibility.

Line 63-64

The drawback of the study is the limited number of the EO’s tested. The screening engaging the higher number of the EO’s with antifungal activity would be more valuable, especially when using such simple test as disk-diffusion essay. Is there any particular reason for choosing such limited number of EO’s?

Line 74-90

In the disk-diffusion assay the negative control is missing. Although the MIC50 and MFC for FLC are evaluated, it should be tested in the assay together with the EO’s.

Discussion

In the discussion are discussed the results of the EOs gas chromatography analysis which are present in the Material and Methods section what presents incorrect manuscript structure.  The results should be presented prior to the discuccsion, as the Table 2.

Secondly, the major part of the discussion (Lines 143-207) is dedicated to these results , and presenting the results of such importance in the Material and Methods section is inexplicable.

The results described in the M&M section in lines as well as the Table 2 should be incorporated in the Results section

228-232.

Table 2 – the % should be explained in the table legend as the % of total EOs sample.

Material and Methods

Line 219 The information about detection is missing. Authors claim that the detection mode was FID, when in the case of this type of detector only the identification on the basis of the standard compounds retentions times is possible. On the other hand the mass range inf the line 227 is given, what suggest that the detector or additional detector was MS. Please specify  the method of detection and characterization of  the compounds or the parameters of the equipment (MS).

On the other hand the FID detection gives more reliable information about quantity of the chemical. Was it used for the quantification or it was based also on the MS analysis.

How the MS/MS analysis was performed? Was the library used or the fragmentation pattern analysis was conducted?

Author Response

The authors appreciate the reviewer's comment. So, the word/concept aromatherapy was modified as recommended and the conclusion of this work has been literally adapted to follow the reviewer's suggestion. Additionally, the negative control (ethanol 70 % (v/v)) was added (Pag 10, line 87-88) and the table legend was modified. Moreover, the choice of the EO was based in clinical evidence provided by our clinician. Although most of the discussion is dedicated to the results in Table 1, the EOs were supplied by the clinician that accompanies our study and therefore we used the characterization included in each oil, it was not made by us (“Analysis of the EOs was carried out by florame® (Saint-Rémy-de-Provence, France) (supplementary material)”). 

Round 2

Reviewer 1 Report

The authors made some revisions to the paper that improve its readability, but I still think that because of the lack of novelty they require additional data. If they could be considered for a shorter format like a note, that would also suffice.

Author Response

The authors agree that the literature provides extensive information on the effect of EOs on Candida spp, at our knowledge, much less information exists on the antifungal effect of EOs either on the biofilm formation, or on mature Candida biofilms, especially on C. auris biofilm (Line 57, Pag. 2). Not forgetting that C. auris is a new species of Candida, therefore, all advances in the direction of promising new therapies, in this case based on natural products, with several advantages such as lower costs, easy access, easier application and less negative impact on the health of the individual is necessary, especially when it involves advanced infection states, such as the biofilm formation.

Reviewer 3 Report

The article has been improved and can be published in the present form. 

Author Response

Thank you for the postive appreciation to our article.

Reviewer 4 Report

I propose to carry out your own analysis (GC/MS) for the studied essential oils and compare it with the literature.

Author Response

The authors are grateful for the reviewer's suggestion. However, the main objective of this work is the evaluation of the antifungal activity of essential oils and not their specific characterization. In addition, the EOs used in the experiments were already used in the clinical practice on alternative medicine and are commercially available, and as such analysed by the responsible company (supplementary material). In fact, this company (Florame) has the advantage of guaranteeing complete traceability, from the producer to the finished product. Thus, laboratory analyses guarantee us essential oils that are rigorously controlled and characterized. Therefore, the authors do not find it strictly necessary to repeat the analysis of essential oils.

Reviewer 5 Report

Unfortunately, the authors did not address the substantial or methodological comments from the previous revision.

Please correct errors in the Manuscript, because in this form, it can not be published in a scientific journal.

I have to rewrite my previous review and stress the objective ERRORS, which were not corrected. 

COMMENET NR 1

The control should be antibiotic, fluconazole or another antifungal agent. It is stated that the activity of natural compounds or mixtures should be compared to drugs when we consider them as future drugs which are supposed to be used internally not externally. Now the activity is compared with disinfectant. 

COMMENT NR 2

The authors should explain why the limited number of Eos was chosen for the preliminary antifungal test in the text of the Manuscript. As stated by the Authors in the response “Moreover, the choice of the EOs was based in clinical evidence provided by our clinician” – if so it can be a part of the Introduction to provide the rationale of the study for future reader, not only for the Reviewer. There are many other EOs of choice with established antifungal activity. e.g., heartwood, marjoram, cinnamon, lemon basil, bay tree, fir, peppermint, pine oil (and many others). Why the bigger number of the EOs was not tested when working with one, drug resistant fungi strain? There is great possibility that more potent EOs were oversight. Were they considered also by the clinician?

COMMENT NR 3

As an active reviewer and scientist, I have never found the description of results in the Materials and Methods section. 

Also the Antibiotics guide for Authors give a clear description of what can be included in Materials and Methods section:

Materials and Methods: They should be described with sufficient detail to allow others to replicate and build on published results. New methods and protocols should be described in detail while well-established methods can be briefly described and appropriately cited. Give the name and version of any software used and make clear whether computer code used is available. Include any pre-registration codes.

Even if the GLC analysis was not made by the Authors (analysis by external entities is a common practice) it is interpreted by them and they should take the responsibility of the results, especially when the results are incorporated into the discussion.

The logical structure of the Manuscript requires describing the results in RESULTS section and methods in MATERIAL AND METHODS section. 

Putting the Table 1 and fragment 286-290 is simply an ERROR, as this manuscript part describe RESULTS.

Please, provide the logical, scientifically correct structure of the Manuscript. 

COMMENT NR 4

As mentioned in the previous review the FID analysis is used for the quantification (or identification on the basis of the standard compounds retention times) when most often MS is used for the qualification. The table 2 gives the information on the identification of VOCs in all EOs, when in line 281 only niaouli oil analysis by the MS detector was used.

I understand that the analysis was made, but in this state the Manuscript do not give sufficient information on 3 essential oils quantitative analysis, when it gives detailed information on the niaouli oil. In my opinion, this kind of incoherence can not be accepted in scientific text and the Authors should gain the information about the other three oil analysis methods from the entity in which they were performed.

Once more i will cite the Guide for the Authors "Materials and Methods: They should be described with sufficient detail to allow others to replicate and build on published results." Now the replication of the analysis in not possible.

Author Response

The authors are grateful for these comments; however, we do not fully agree with the fact that the reviewer says that we did not address his previous comments since we answered everything and also changed things accordingly. Moreover, we do not agree with the aggressive tone of the recommendations.